# Structural basis for the mechanisms of human presequence protease conformational switch and substrate recognition

Wenguang G. Liang[1], Juwina Wijaya[2], Hui Wei [3], Alex J. Noble [3], Jordan M. Mancl[1], Swansea Mo[1], David Lee [4], John V. Lin King[5], Man Pan [6], Chang Liu[6], Carla M. Koehler[2], Minglei Zhao [6], Clinton S. Potter[3], Bridget Carragher [3], Sheng Li [4] & Wei-Jen Tang [1]✉

Presequence protease (PreP), a 117 kDa mitochondrial M16C metalloprotease vital for mitochondrial proteostasis, degrades presequence peptides cleaved off from nuclear-encoded proteins and other aggregation-prone peptides, such as amyloid β (Aβ). PreP structures have only been determined in a closed conformation; thus, the mechanisms of substrate binding and selectivity remain elusive. Here, we leverage advanced vitrification techniques to overcome the preferential denaturation of one of two ~55 kDa homologous domains of PreP caused by air-water interface adsorption. Thereby, we elucidate cryoEM structures of three apo-PreP open states along with Aβ- and citrate synthase presequence-bound PreP at 3.3–4.6 Å resolution. Together with integrative biophysical and pharmacological approaches, these structures reveal the key stages of the PreP catalytic cycle and how the binding of substrates or PreP inhibitor drives a rigid body motion of the protein for substrate binding and catalysis. Together, our studies provide key mechanistic insights into M16C metalloproteases for future therapeutic innovations.

[1] Ben-May Department for Cancer Research, The University of Chicago, Chicago, IL, USA. [2] Department of Chemistry and Biochemistry, University of California Los Angeles, Los Angeles, CA, USA. [3] National Resource for Automated Molecular Microscopy, Simons Electron Microscopy Center, New York Structural Biology Center, New York, NY, USA. [4] Department of Medicine, University of California San Diego, La Jolla, CA, USA. [5] Department of Physiology, University of California, San Francisco, San Francisco, CA, USA. [6] Department of Biochemistry and Molecular Biology, The University of Chicago, Chicago, IL, USA. ✉email: wtang@bsd.uchicago.edu

Mitochondria are vital to cellular metabolism, homeostasis, and stress responses[1,2]; their defects are linked to a plethora of neurodegenerative diseases[3]. Assembly of mitochondria requires the coordinated action of protein import into mitochondria, coupled with processing and proteolysis pathways[4,5]. Most imported proteins contain a presequence (also known as a mitochondrial targeting sequence) at the N-terminus[4]. Upon entrance into mitochondria, presequences are cleaved off by mitochondrial processing peptidase and in some cases further cleaved by mitochondrial intermediate proteases[4,5]. Presequences are rich in hydrophobic and positively charged residues and highly toxic to mitochondria if left to accumulate[4,5]. Presequence protease (PreP) is a ubiquitously expressed, 117 kDa M16C clan zinc metalloprotease that localizes to the mitochondria matrix and cleaves presequence peptides into non-toxic pieces[4,6,7]. PreP also degrades amyloid β (Aβ), the cleavage product of the amyloid precursor protein (APP) that is linked to the progression of Alzheimer's disease and is imported into mitochondria, particularly synaptic mitochondria[6–9]. A homozygous Pitrm1 (which encodes PreP) knockout in mice displays embryonic lethality[10]. Furthermore, genetic defects in PreP are linked with human neurological disorders, e.g., cognitive disability/impairment and cerebellar atrophy[10–12].

Proteomes are maintained in a healthy state by four main processes, autophagy, chaperones, ubiquitination/proteasomal degradation, and a cohort of proteases that degrade potentially cytotoxic peptides[13,14]. Aggregates of such peptides include amyloidogenic species that are highly cytotoxic and are associated with human neurodegenerative diseases, e.g., Alzheimer's and Parkinson's disease, and non-neuropathic systemic amyloidoses[15–17]. Aβ is a key initiating factor in Alzheimer's disease and its accumulation is caused by the imbalance between Aβ production and clearance[17]. Aβ is the proteolytic product of the amyloid precursor protein (APP), which belongs to a small gene family including APP-like proteins that have shared functions in CNS development, synapse formation, brain injury, and neuroprotection[18]. Only through the processing of APP, a recently evolved gene within the APP family can generate Aβ and no dedicated protease has evolved for Aβ clearance[18]. A handful of proteases out of 569 human proteases can effectively degrade monomeric Aβ[6,19,20]. Aβ-degrading proteases have a broad subcellular distribution, e.g., extracellular milieu (insulin-degrading enzyme (IDE), matrix metalloprotease 2 (MMP2), MMP9); mitochondria (PreP), intracellular vesicles (IDE), lysosomes (cathepsin B), the plasma membrane (neprilysin (NEP) and endothelin converting enzyme 1 and 2 (ECE1/2), plasmin, IDE), and the cytosol (IDE, acyl-peptide hydrolase (APEH)). Together these enzymes enable better control of Aβ levels at all intra- and extra-cellular locations where Aβ has been detected.

The formation of aggregates of amyloid peptides such as Aβ is at least a two-step process; the first is a slow, stochastic, and reversible nucleation to form small amyloid peptide seeds followed by the elongation of seeds into large amyloid fibrils, which is faster and mostly irreversible[21,22]. Monomeric amyloid peptide fuels the forward progression of both steps; thus, Aβ-degrading proteases that recognize and degrade monomeric amyloid peptides prevent the formation of amyloid fibrils[19]. Of the Aβ-degrading proteases, PreP belongs to the chamber-containing protease (crypt-containing peptidase, or cryptidase) family that uses a sizable catalytic chamber to engulf, unfold, and degrade their substrates[23]. Other cryptidases include IDE (M16A clan) and M13 metalloproteases, e.g., NEP and ECE1/2[23]. Aβ-degrading proteases also can selectively degrade other amyloid peptides which are highly diverse in sequence and structure. This raises a major question: how do such proteases selectively degrade amyloid peptides over non-amyloidogenic peptides?

Crystallographic analyses reveal that PreP and related M16C metalloproteases are composed of ~55 kDa homologous N- and C-terminal domains (PreP-N and PreP-C, respectively), which are connected by an extended helical hairpin[6,7]. PreP-N and PreP-C, in the closed state of PreP, form an enclosed catalytic chamber to entrap and degrade monomeric amyloidogenic peptides, thereby preventing the formation of toxic aggregates[6]. However, the structure of the catalytic chamber in the PreP closed state precludes the capture of its substrates such as Aβ, or the release of its reaction products, which are key steps in the PreP catalytic cycle (Supplementary Fig. 1). To date, no structure of an open state of the M16C clan of metalloprotease has been reported. Solution scattering studies indicate that human PreP in solution is mostly in a closed or partially closed state[6]. However, the structural basis for the closed-open transition that allows for substrate capture and release of proteolysis products, in addition to substrate-induced transition from open to the closed state, remains elusive (Supplementary Fig. 1). Here, we integrate cryoEM, crystallography, size-exclusion chromatography (SEC) coupled small-angle X-ray scattering (SAXS), hydrogen–deuterium exchange (HDX)–mass spectrometry (MS), site-specific mutagenesis, and chemical biology approaches to elucidate the structural basis of the open-closed transition and the mechanism of substrate recognition for human PreP.

## Results

**Solution of multiple open and substrate-bound closed structures by cryo-EM.** Advances in cryo-electron microscopy (cryoEM) allow the structural determination of conformational states recalcitrant to crystallography[24–27], thus we used cryoEM to examine the conformational states of PreP in the absence of substrate (apo-PreP). We first used differential scanning fluorimetry (DSF) to optimize the unfolding and dissociation enthalpy of human PreP (Supplementary Fig. 2)[28]. Apo-PreP grids were then prepared using a Vitrobot and a dataset of 2,626 micrographs was collected at 300 kV on a Titan Krios at various ice thicknesses (Table 1 and Supplementary Figs. 3 and 4). During processing, we observed that the predominant classes contained particles only half the expected size of PreP (Fig. 1a and Supplementary Fig. 3). Following the 3D classification of 411,000 particles, four classes were obtained (Supplementary Fig. 3). The two major classes, comprising ~208,000 and ~118,000 particles, displayed an intact PreP-N domain and a denatured PreP-C domain, which both were refined to 4.2 Å (Fig. 1b and Supplementary Figs. 4 and 5). The third class, comprised of 50,000 particles and refined to 4.5 Å, was found to contain full-length PreP adopting a partially open conformation that we designate pO (Fig. 1b and Supplementary Fig. 3). The final class, of ~34,000 particles and refined to 5.3 Å, comprised of the intact PreP-N and partially denatured PreP-C that was in pO state (Supplementary Fig. 3).

It has previously been demonstrated that more than 90% of particles derived from diverse proteins or protein complexes are adsorbed to the air–water interface (AWI) in cryoEM[29]. Furthermore, extensive studies have shown that many proteins are denatured rapidly upon exposure to the AWI[30]. Therefore, we hypothesized that the extensive denaturation of the PreP-C domain described above resulted from denaturation at the AWI. To explore this possibility, we used fiducial-less cryo-electron tomography (cryoET) to examine the distribution of PreP particles within the vitrified ice of the same cryoEM grids. This analysis of Vitrobot-prepared grids revealed that nearly all PreP particles were adsorbed to the AWI (Fig. 1c, Supplementary Fig. 6, and Movie 1). Approximately, ~88% of particles had half of the anticipated size in our cryoET analysis, consistent with

**Table 1 CryoEM data collection, refinement, and validation statistics.**

| Data collection and processing | Apo-PreP | | | Aβ-bound PreP | CS27-bound PreP |
|---|---|---|---|---|---|
| Conformations | Partial Open 1 (pO1) | Partial Open 2 (pO2) | Open (O) | Partial Closed (pC) | Partial Closed (pC) |
| Magnification | 130,000 | | | 130,000 | 130,000 |
| Voltage (kV) | 300 | | | 300 | 300 |
| Electron exposure (e−/Å²) | 62.36 | | | 66.39 | 66.14 |
| Defocus range (µm) | −2.5 to −1.5 | | | −2.0 to −1.2 | −2.5 to −1.5 |
| Pixel size (Å) | 0.856 | | | 0.855 | 0.855 |
| Symmetry imposed | C1 | | | C1 | C1 |
| Initial particle images (no.) | 356,754 | 356,754 | 334,473 | 213,544 | 1,799,857 |
| Final particle images (no.) | 130,572 | 139,127 | 93,593 | 174,537 | 330,536 |
| 3D-FSC sphericity | 0.86 | 0.871 | 0.861 | 0.837 | 0.878 |
| Map resolution (Å) | 3.7 | 3.9 | 4.0 | 3.3 | 4.6 |
| FSC threshold | 0.143 | 0.143 | 0.143 | 0.143 | 0.143 |
| EMDB | EMD-22278 | EMD-22279 | EMD-22280 | EMD-22281 | EMD-22282 |
| **Refinement** | | | | | |
| Map sharpening B factor (Å²) | −56 | −88 | −92 | −40 | −292 |
| Model composition | | | | | |
| Protein residues | 966 | 966 | 965 | 972 | 966 |
| Total atoms | 7704 | 7704 | 7697 | 7769 | 7711 |
| Substrate | – | – | – | 51 | – |
| B factors (Å²) | | | | | |
| Protein | 111 | 132 | 102 | 90 | 136 |
| Substrate | – | – | – | 88 | – |
| R.m.s deviations | | | | | |
| Bond length (Å) | 0.009 | 0.006 | 0.005 | 0.004 | 0.006 |
| Bond angle (°) | 1.080 | 1.086 | 1.081 | 0.668 | 0.976 |
| Ramachandran (%) | | | | | |
| Favored | 97.61 | 97.19 | 97.19 | 94.91 | 95.84 |
| Allowed | 2.39 | 2.81 | 2.81 | 5.09 | 4.16 |
| Outliers | 0.00 | 0.00 | 0.00 | 0.00 | 0.00 |
| Validation | | | | | |
| MolProbity score | 1.17 | 1.41 | 1.32 | 1.98 | 2.15 |
| Poor rotamers (%) | 0.00 | 0.23 | 0.35 | 0.00 | 0.23 |
| Clash score | 2.88 | 4.98 | 3.80 | 13.06 | 23.48 |
| PDB | 6XOS | 6XOT | 6XOU | 6XOV | 6XOW |

predominant 3D classes (80%) having a denatured PreP-C domain (Fig. 1c and Supplementary Fig. 6A). Interestingly, while PreP-N and PreP-C share a highly similar structure, PreP-N has an additional β-hairpin (Supplementary Fig. 6B). This β-hairpin extends from an α-helix that binds the catalytic zinc ion and interacts with the α-helical hairpin that links PreP-N and PreP-C. This structure likely makes PreP-N more stable than PreP-C (Supplementary Fig. 6B). Together, our data indicate that PreP-C is preferentially denatured during the vitrification process, either by the repetitive exposure to AWI and/or shear force caused by grid blotting (Supplementary Fig. 6C)[31].

We hypothesized that if the amount of time the sample spent on the grid prior to vitrification (dwell time) could be significantly reduced, PreP denaturation would likewise be reduced. Spotiton, a novel method of vitrifying samples using a piezoelectric dispensing head to deliver sample droplets onto a self-blotting nanowire grid, has been shown to significantly reduce the dwell time of particles at the AWI prior to vitrification[31–34]. We employed this technique to prepare grids using a 133 ms dwell time (compared to 1–2 s for Vitrobot) by chameleon[35], a commercial version of Spotiton developed by SPT Labtech. An apo-PreP dataset of 3012 micrographs was processed to yield about 363,000 particles from these grids, which adopted well-defined 3D classes. Following 2D and 3D classification, all classes were found to contain full-length PreP particles, and no denaturation was observed (Fig. 1d, e, Table 1, and Supplementary Figs. 7 and 8). 3D classification of PreP particles revealed

three distinct open state structures of PreP. They were refined to an open state (O) and two partially open states (pO1 and pO2) with resolutions of 4, 3.7, 3.9, respectively. Structural models of these three states were then built and refined (Table 1, Supplementary Figs. 5B–D, 8, and Movie 2). Two substrate-bound PreP cryoEM structures were also determined. We also optimized the conditions for determining the structure of PreP in the presence of a five-fold molar excess of Aβ 1–40 by DSF. Grids were prepared by the chameleon, and a dataset of 3483 micrographs was processed to yield about 175,000 particles from a well-defined 3D class that was refined to a partially closed (pC) state of PreP at 3.3 Å resolution (Fig. 1f, Table 1; Supplementary Figs. 5E, 9, and 11). A similar approach was used to obtain a map for PreP in complex with a model presequence peptide derived from human citrate synthase (27 aa long, CS27)[36], resulting in pC state PreP at 4.6 Å resolution (Fig. 1g, Table 1; Supplementary Figs. 10 and 11). Structural models of substrate-bound PreP were then built and refined (Table 1, Supplementary Figs. 5E, 11, and Movie 3). We define the substrate-bound PreP cryoEM structures as pC state that is slightly more open than the closed (C) state crystal structures of PreP solved in the presence or absence of Aβ[6].

**Structural analysis of apo- and substrate-bound PreP reveal key states in PreP catalytic cycle and the molecular basis for substrate recognition.** Comparison of the five cryoEM structures

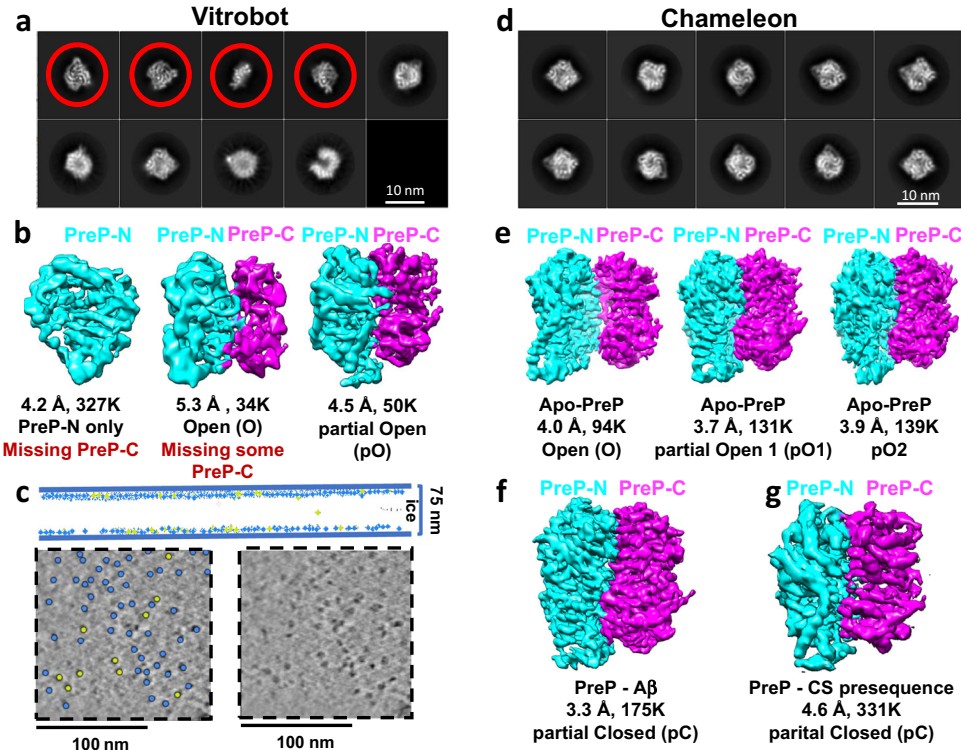

**Fig. 1 CryoEM analysis of PreP.** **a** 2D classification, **b** 3D classification, and **c** cryoET analysis of PreP alone using grid prepared by Vitrobot. Top, Schematic diagrams of the average ice thickness (solid blue lines), and particle distribution in the ice. Almost all particles are on the AWI (770 particles), and only one full particle is not absorbed into the AWI. Bottom, Comparison of an enlarged slice of tomograms with particles' labels (left) and without particles labels (right). PreP-N and PreP-C are colored in cyan and magenta, respectively. **d** 2D and **e** 3D classification of PreP alone using grid prepared by chameleon. **f**, **g** 3D classification of PreP in complex with Aβ (**f**) and citrate synthase (CS) presequence (**g**).

derived from the apo- and substrate-bound PreP reveals three key conformational states based on the degree of opening of the catalytic chamber: open (O), partially open (either pO1 or pO2), and partially closed (pC). Apo-PreP has three states, O, pO1, and pO2 that are distinct from each other with the two pO states being slightly more open than the pC or C states (Fig. 2a, Supplementary Fig. 12, and Table 1). Aβ- and CS27-bound PreP structures in the pC state are nearly identical to each other and are slightly more open than crystal structures of apo or Aβ-bound PreP[6] (Fig. 2a, Supplementary Fig. 12 and Table 1). Similar to the Aβ-bound PreP crystal structure[6], extra-densities at the catalytic cleft and hydrophobic sites away from catalytic zinc ion within the catalytic chamber of PreP were found when Aβ was present, confirming key substrate-binding sites of PreP. Of these structures, the pO, pC, and C states of PreP have a small variation in the degree of opening, making their chamber inaccessible to substrate binding. Thus, only the PreP open state has a large enough opening to capture its peptide substrates and release the proteolytic products.

The PreP open state differs from the rest of PreP states in two major ways. The first is mediated by the rigid body displacement between PreP-N and PreP-C domains, whereby the two halves of this chamber-containing protease open up, similar to a clamshell (Fig. 2a–c, Supplementary Fig. 12, Movie 4, and Table 1). The displacement results in a difference in the distance, angle, and contacts between these domains. Most noticeably, both the distance and angle between PreP-N and PreP-C in the PreP open state is substantially larger than the rest of PreP states while the buried surface between PreP-N and PreP-C is much reduced compared to the others. The displacement between PreP-N and PreP-C is most likely driven by the entropically favorable rigid body motion. We also observed major conformational

rearrangements in two additional regions, which we term switch A (aa 174–225) and switch C (aa 506–550) (Fig. 2a–c; Supplementary Fig. 12 and Movie 4). Switch A contains two α-helices that have residues for the binding of substrate and catalytic zinc ion (Fig. 2a). The helix-turn-helix motif of the switch C region joins PreP-N and PreP-C and makes extensive contacts with an extended β-hairpin within the long α-helix of the switch A region, allowing the switch A and C region move jointly with the displacement between PreP-N and PreP-C (Fig. 2a).

3D classification revealed that most PreP particles (~74%) were in the partially open (pO) states, indicating that PreP in the absence of substrate prefers to be in a state inaccessible to substrate binding. We then used size exclusion chromatography (SEC) in line with SAXS to further assess how the distribution of PreP conformational states in solution is influenced by the presence of Aβ and presequence from citrate synthase (CS27) under physiological buffer conditions (Fig. 2d, Supplementary Table 2 and Fig. 13). SAXS is a highly effective technique to eliminate structural models that do not produce calculated scattering patterns that fit the experimental scattering profile[37]. The direct coupling of SEC just prior to SAXS analysis reduces large aggregates that contaminated our previously reported SAXS profile of PreP[6]. Consistent with the cryoEM data, the SEC-SAXS data confirms that apo-PreP in solution also prefers to adopt the pO state (72% based on OLIGOMER and 90% based on ensemble optimization modeling (EOM)) (Fig. 2, Supplementary Table 2 and Fig. 13)[38–40]. The fact that PreP in solution prefers the pO state rather than the open state is logical because the transition from the pO states to O state loses substantial buried surface (650–940 Å$^2$) and a network of hydrogen bonds and salt bridges, and thus is energetically unfavorable (Supplementary Table 1). The presence of CS27 significantly reduced the $R_g$ value from 31.6

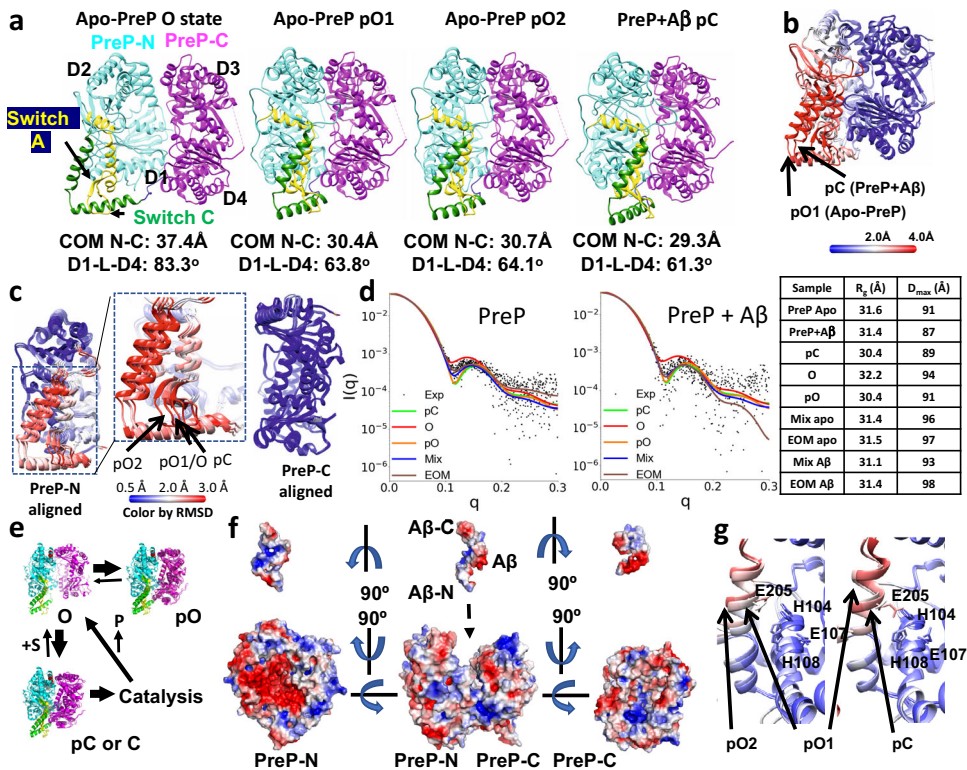

**Fig. 2 Comparison of PreP structures. a** Four key cryoEM structures of PreP. Conformations of PreP open (O), partial open state 1 and 2 (pO1 and pO2), and Aβ-bound partial closed states (pC) are shown in the ribbon. The distance between the center of mass (COM) of PreP-N and PreP-C and the angle for COM of D1, aa 562-564 at switch C, and D4 domain are listed below each conformation. PreP-N, PreP-C, switch A, and switch C domains are colored in cyan, magenta, yellow, and green, respectively and the color scheme is used throughout the figure. **b** Alignment of pO1 and pC state of PreP colored by RMSD as indicated. **c** Alignment of PreP-N and PreP-C in four distinct conformational states of PreP colored by RMSD as indicated. **d** SEC-SAXS analysis of PreP in the presence and absence of Aβ. Mix denotes the scattering profile for a mixed population of known PreP structures as calculated in OLIGOMER. EOM denotes the scattering profile resulting from ensemble optimization modeling. While the major classes resulting from EOM showed high similarity to the states observed in our cryoEM analysis, a minor class that opened to a degree beyond that which has been previously observed experimentally were used in both PreP alone and PreP + Aβ conditions. As such minor classes also inflated the $D_{max}$ compared to experimental conditions, the OLIGOMER analysis was found to produce a better fit to the experimental data than the EOM analysis. See Supplementary Fig. 13 for more details. **e** Model of PreP catalytic cycle. S is a substrate and P is proteolytic products. **f** Structural basis of PreP open state primed to capture Aβ by size and charge complementarity. The charge distribution is calculated using PYMOL APBS plugin. The negative and positive charged surfaces are shown in red and blue, respectively. **g** Structural comparison of the catalytic site of pO and pC states colored based on RMSD.

to 30.8 Å. Given the fact that the predicted SAXS profiles and $R_g$ values of pO and pC are nearly identical, our analysis reveals that the presence of CS27 leads PreP to be almost entirely in the closed state (100% and 91% based on OLIGOMER and EOM, respectively) (Supplementary Fig. 13). However, the presence of Aβ only slightly reduced the $R_g$ and $D_{max}$ values, which slightly increases the percentage of closed states (from 72% to 80% using OLIGOMER and from 50 to 54% using EOM). This is consistent with only a subtle conformational switch occurring between pO and pC states (Fig. 2d, Supplementary Fig. 13, and Table 1), whereby substrate-binding promotes domain closure. It is worth noting that EOM consistently generated a minor class of PreP that is much more open than the observed cryoEM open state of PreP (Supplementary Fig. 13). Thus, the analysis using OLIGO-MER likely represents the more realistic estimation.

Together, our data lead to the hypothesis that PreP undergoes the following conformational switch during the catalytic cycle (Fig. 2e): PreP is predominantly in the partially open state at the resting condition. The transition from the pO states to O states allows the capture of the substrate, and thus is a key state in the catalytic cycle. Upon opening, peptides that are rich in positively charged residues are attracted to the negatively charged, catalytic chamber of PreP-N which can further select for its substrates

based on their size and conformational compatibility within the chamber (Fig. 2f)[6]. The catalytic site of PreP undergoes a minimal conformational change, and thus is poised to carry out proteolysis (Fig. 2g). After proteolysis, the closed to open transition allows the release of proteolytic products to initiate the next catalytic cycle.

**The mechanism for the conformational switches between PreP open and closed states in the presence and absence of substrates.** The comparison of PreP cryoEM structures reveals the molecular basis for the equilibrium between the partially open and open states in the absence of substrate. PreP has three regions that undergo substantial conformational switches, defined as switch A–C (Fig. 3a). As discussed above, switch A and C move together with rigid body displacement between PreP-N and PreP-C (Figs. 2a and 3a). The switch A and C regions in the pO states have lower resolution and higher thermal B factors than the rest of PreP structures (Fig. 3b). In comparison, the resolution and thermal B factors of these regions in the open and substrate-bound states are not profoundly different from the rest of the protein (Fig. 3b). Together, this is indicative of high conformational dynamics within these regions. As switch A contains

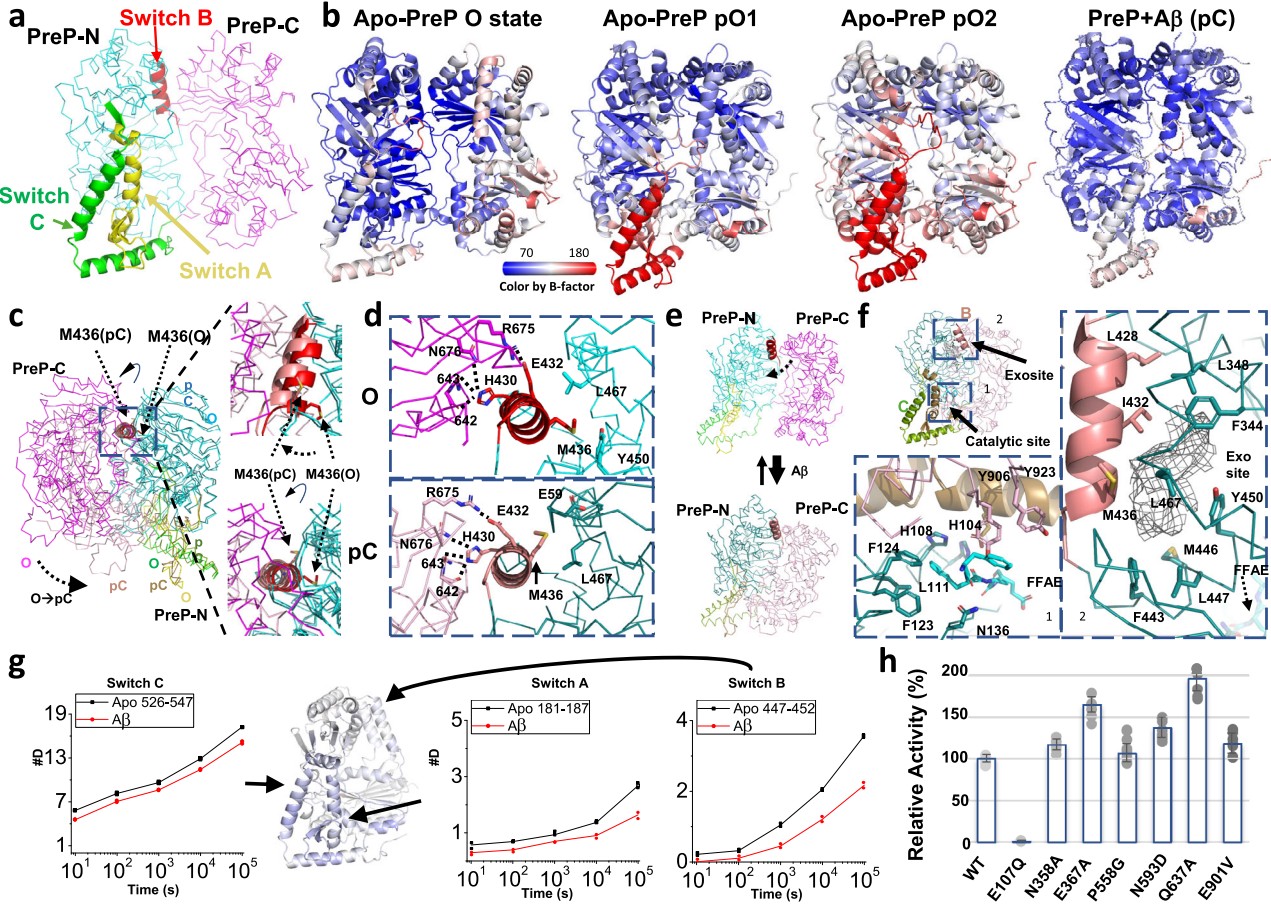

**Fig. 3 Conformational switch of PreP. a** Three switch regions of PreP. PreP-N, PreP-C, switch A, switch B, and switch C domains are colored in cyan, magenta, yellow, red, and green, respectively and the color scheme is used throughout figures. **b** Four cryoEM PreP structures colored by B-factors as indicated. **c** Comparison of PreP open and partial closed state to show the rotation of switch B helix that governs the open-closed transition. **d** The detailed interactions of switch B region in the open (top) and partial closed (bottom) states of PreP. **e** Model depicting how the rotation of switch B region governs the preferred parital closed state in the absence of substrate and the interaction of hydrophobic residues in the substrate, e.g., Aβ induces the open to closed transition of PreP. **f** The interaction of Aβ with the catalytic site (box 1) and exosite (box 2) of PreP. The key residues for Aβ binding and those that form exosite are shown in stick and labeled. The density found in Aβ bound PreP structure is shown in the mesh. The [19]FFAE[22] in Aβ is colored in cyan. **g** Differential HDX between PreP in the absence and presence of Aβ mapped on the PreP-N structure. Differences in the average HDX are represented as percent change and colored with blue being slower exchange with Aβ and with red being faster. The differences in HDX from $n = 2$ technical replicates are shown. **h** Relative catalytic activities of PreP mutants that have point mutations at residues residing at the interface between PreP-N and PreP-C. The average relative catalytic activities were shown in scale bar ±SD, and the original data $n = 8$ was overlaid on the scale bar. Source data are provided as a Source Data file.

residues involved in substrate peptide binding and catalytic zinc ion coordination, such high dynamics would render the pO state catalytically incompetent. The presence of substrate stabilizes switch A, and thus, the residues for substrate binding and catalysis, enabling the catalytic reaction. This suggests that PreP uses substrate-assisted stabilization as a mechanism for substrate catalysis. This is because amyloid peptides have a high propensity to unfold and form a β-strand, which can then form the cross-β-sheet with other amyloid peptides. The catalytic cleft of PreP selectively binds the β-strand of substrate peptide after such peptide is unfolded inside the catalytic chamber. Only peptides that tend to unfold and form the β-strand can stabilize PreP's catalytic cleft, leading to catalysis.

PreP switch B (aa 421–436, within PreP-N) represents a rotation of an α-helix when the PreP open state is compared with the rest of conformational states (pO, pC, or C) (Fig. 3c, d; Supplementary Movie 5). The rotation of switch B is particularly noticeable at residue M436, which rotates ~60° to transit through the hydrophobic pocket formed by Y450 and L467 (Fig. 3c, d).

This allows a rigid body rotation of PreP-C in relationship to PreP-N (Figs. 2e and 3e). Such a rotation maintains the interactions between switch B and PreP-C, e.g., the contacts of E432 with R675 and that of H430 with N676 and the main chains of aa residues 642 and 643. Thus, there should be a minimum energy barrier for such rotation to allow the rapid shift between PreP partially open and open states driven by the entropically favorable rigid body motion between PreP-N and PreP-C.

The rotation of switch B also offers the molecular basis for PreP's substrate-induced conformational switch and substrate selectivity. Switch B has hydrophobic residues L428 and I432 that, together with the surrounding residues, form a substrate-binding exosite distal to the catalytic zinc ion (~28–33 Å away) (Fig. 3f and Supplementary Movie 5). This exosite is highly hydrophobic and has been observed to bind Aβ residues that are away from the preferred cleavage sites of Aβ (aa 19/20 or aa 20/21) (Fig. 3f)[6]. The interaction of the exosite with the hydrophobic residues of substrate would favor the closed state and thus promote the open to closed transition (Fig. 3e). Furthermore, this can in part

explain why PreP prefers to degrade peptide substrates that are rich in hydrophobic residues, e.g., presequences and Aβ.

Peptide amide HDX-MS is a powerful tool to probe protein conformational dynamics because it allows evaluation of comparative solvent accessibility throughout the protein[37,41–45]. We used HDX-MS to test the hypothesis that the conformationally dynamic switch regions are stabilized by the binding of substrates (Aβ and CS27) (Supplementary Table 3 and Figs. 14–17, and Supplementary Data 1–2). Both 2.7 Å resolution crystal[6] and 3.3 Å resolution cryoEM structures of Aβ-bound PreP showed that several discrete regions within the catalytic chamber of PreP are involved in the recognition of presequence and Aβ. Residues from both PreP-N (aa F123, F124, L127, 135–139, and M206) and PreP-C (aa R900 and Y906) form a catalytic cleft to bind aa18–22 of Aβ. PreP prefers to degrade peptides that are rich in hydrophobic and positively charged residues. The recognition of hydrophobic and positively charged residues is mediated by a hydrophobic pocket formed by L428, I432, F344, L465, L60, F443, L447, and Y450 and a negatively charged pocket formed by D212, E213, D377, and D716, respectively. As expected, reduced HDX was observed in segments in both PreP-N and PreP-C that are involved in the substrate binding (e.g., aa 115–140, aa 166–175, and aa 893–921) (Supplementary Table 3 and Figs. 14–17, and Supplementary Data 1–2). Consistent with our SEC-SAXS data that the binding of substrates (both Aβ and CS27) promotes PreP to transition from the open to a closed state, we also observed the reduced HDX at the interface between PreP-N and PreP-C (e.g., aa 634–641, aa 705–722, and aa 922–933) in the presence of substrate. In addition to the expected changes in substrate binding and PreP-N/PreP-C interface, we observed reduced HDX of switch A–C regions when two different PreP substrates, Aβ and CS27 were present, which confirmed our prediction (Fig. 3g and Supplementary Figs. 14–17).

To probe how the interface between PreP-N and PreP-C controls the equilibrium between the pO and O states, we carried out structure-guided mutagenesis in this region at positions predicted to weaken the interaction between PreP-N and PreP-C. We found that two point mutations, D367A and Q637A modestly increased the catalytic activity of PreP (Fig. 3h). Thus, destabilization of this interface enhances, rather than diminishes, PreP's enzymatic activity, presumably through increasing the ease with which PreP can transition through the key conformational states of its catalytic cycle.

**Mechanism of PreP inhibition by MitoBlocker-60.** At the PreP-N and -C interface, we observed an intriguing overlap between a key conformational switch (B) and a key substrate-binding site, the exosite, whose functional role in substrate-binding and catalysis is largely unexplored. To further define the conformational dynamics at this interface, we exploited the findings of an in vitro high-throughput screen that identified MitoBlocker-60 (MB60, 1-(diphenylmethyl)-4-(3-methyl-4-nitrobenzoyl)piperazine) as a potent inhibitor of PreP (Fig. 4a). MB60 inhibited the degradation of Aβ by PreP with an $IC_{50} = 200$ nM and triggered mitophagy under mitochondrial stress[46]. A previous CRISPRi screen showed that PreP was essential for the robust cell proliferation of human K562 leukemia cells[47]. Consistent with this notion, MB60 potently blocked cell proliferation in a dose-dependent manner without inducing cell death (Fig. 4a). To understand the mechanism of inhibition, we co-crystallized MB60 with human PreP. The structure of MB60-bound PreP at 2.3 Å resolution reveals an unexpected binding mechanism of MB60 (Fig. 4b, Table 2, and Supplementary Fig. 18). In the presence of MB60, PreP exists in a closed conformation that is nearly identical to structures of substrate-free and Aβ-bound PreP (RMSD = 0.15 Å and 0.31 Å, respectively)[6]. Within the catalytic

chamber, two MB60 molecules wrap around each other to make intimate interactions to bury 313 Å² and bind the distinct pockets at the PreP exosite via various contacts to bury 808 and 658 Å² surfaces of MB60-a (pink) and MB60-b (yellow), respectively (Fig. 4b, c). For MB60-a, two phenyl groups bind a hydrophobic pocket in close contact with L60, F344, I432, M446, L447, and L467. The carbonyl group of MB60-a forms hydrogen bonds with waters coordinated by the carbonyl group of M446 and the hydroxyl group of Y383. The piperazine group of MB60-a forms a hydrogen bond with water coordinated with the side chain of Q435. The nitro group of MB60-a forms a hydrogen bond with the main chain of G382 and a cation-π interaction with Y383. For MB60-b, two phenyl groups interact with the hydrophobic pocket formed by I337, A343, F344, I451, and L464 while the nitro group of MB60-b forms a salt bridge with K431, thereby favoring the pO state. The finding that MB60 targets PreP's exosite provide strong evidence that this site plays a critical role in PreP catalysis.

To explore how MB60 affects the substrate-binding and conformational dynamics of PreP, we first used SEC-SAXS and showed that MB60 did not induce obvious changes in the SAXS profile (Fig. 4e and Supplementary Figs. 13 and 14). The effect of MB60 on the conformational dynamics of PreP was then explored by the differences between amide H/D exchange profiles of PreP alone and PreP in the presence of MB60. Most exchanges were unchanged (Fig. 4f and Supplementary Figs. 14 and 15). Of a few regions that showed a noticeable reduction in H/D exchange, the peptides around MB60 binding pockets were most prevalent. These included residues 355–391 and 418–477 (Fig. 4f and Supplementary Fig. 19). This supports the notion that MB60 in solution binds the hydrophobic pockets revealed by our MB60-bound PreP structure. While reduced exchange in the switch B region is expected as it is a part of the MB60 binding site, segments in switch A and switch C regions (aa 207–217 and aa 506–525, respectively) that are away from the MB60 binding site also displayed reduced H/D exchange (Fig. 4f and Supplementary Fig. 19). This is consistent with our model that in the absence of MB60, the switch A and C regions of PreP undergo dynamic motion. Moreover, that binding of MB60 to the exosite stabilizes such motion suggests that conformational changes in the exosite are functionally coupled to those in switches A and C. It is worth noting that, similar to the binding of substrates, Aβ and CS27, segments in PreP-C that are near the catalytic site also had the reduced exchange (e.g., aa 703–723, and aa 898–923, Fig. 4f and Supplementary Fig. 19). This is consistent with our model that the binding of MB60 to the switch B region can further stabilize the pO states, leading to a better interaction between PreP-N and PreP-C.

Together, our structural and HDX-MS analysis explains how MB60 prevents degradation of Aβ and other substrates. MB60 is too large to enter the catalytic chamber of PreP when PreP is in the pO state, the dominant state in the solution. Upon entrance into the catalytic chamber via the PreP open state, MB60 promotes the rotation of switch B, which in turn promotes the open to closed transition. Such interaction should disfavor the pO to O transition, preventing Aβ from accessing the catalytic chamber. Furthermore, MB60 also blocks Aβ from binding to PreP exosite even after Aβ enters the PreP catalytic chamber. The close interactions of PreP with both phenyl groups of MB60 also explain the structure-activity relationship of MB60[46]: loss of one phenyl group increased the $IC_{50}$ value five-fold while the relocation of the nitro group within the tolyl group of MB60 increased the $IC_{50}$ value 25-fold.

## Discussion

Major advances in instruments, techniques, and methods have fueled a "resolution revolution", making single-particle cryoEM a

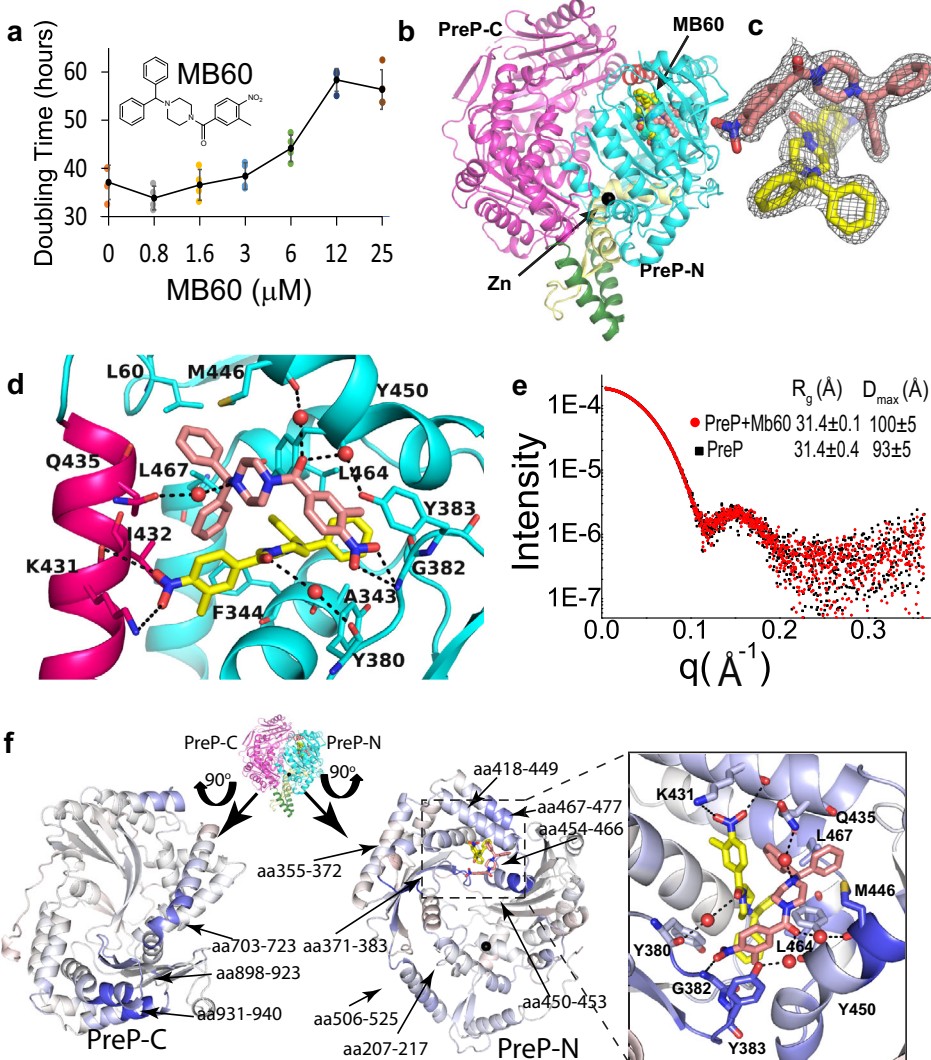

**Fig. 4 Structural analysis of PreP in complex with small molecule inhibitor, MB60. a** The effect of MB60 on the doubling time of K562 Leukemia cells. The chemical formula of MB60 is shown. Error bar is SD from $n = 0.5 \times 10^4$ cells (initial cell numbers) over 4 independent experiments. Source data are provided as Source Data files. **b** Overall structure of MB60-bound PreP (PDB code: 4RPU) PreP is depicted in a ribbon representation. PreP-N, PreP-C, switch A, switch B, and switch C domains are colored in cyan, magenta, yellow, red, and green, respectively and the color scheme is used throughout figures. The carbons of two MB60 molecules are colored in yellow and pink while N and O atoms are in blue and red, respectively. The catalytic zinc ion is in gray. **c** 2mFo–DFc omit map of MB60 to depict the closed contact between two MB60 molecules at the exosite. The map was contoured to $1\sigma$. **d** Detailed interactions of two MB60 with PreP side chains within the exosite. **e** SEC-SAXS scattering profile of PreP in the presence or absence of MB60 (dotted lines). Theoretical scattering profiles of open and closed PreP (solid lines) were modeled and calculated by CRYSOL. **f** Differential HDX between PreP in the absence and presence of MB60 mapped on the MB60 bound PreP crystal structure. Differences in the average HDX are represented as percent change and colored with blue being slower exchange with MB60 while with red being faster.

powerful structural determination technique that rivals macromolecular crystallography[24–27]. However, rapid protein denaturation in the thin film generated during the grid-making process, due to the high surface area to volume ratio at the AWI, represents a major obstacle in identifying the suitable condition to vitrify protein sample for cryoEM analysis[30]. Our cryoET analysis reveals that PreP, a 117 kDa, monomeric enzyme with homologous 55 kDa N- and C-domains, is preferentially absorbed to the AWI. Such exposure likely caused preferential denaturation of the PreP-C domain. The ability of the PreP-N domain to withstand exposure to the AWI, despite its high degree of structural homology to the PreP-C domain, is likely due to enhanced stability granted by the lock formed by switch A and switch C regions (Supplementary Fig. 6). The combination of self-blotting nanowire grids and piezo dispensing utilized by the chameleon, the

commercial version of Spotiton, eliminated the paper blotting step and significantly reduced the time for PreP to interact with the AWI[31–35]. This led to the successful determination of apo- and substrate-bound PreP cryoEM structures. Thus, PreP is a compelling case study that demonstrates how reduced vitrification times can be used to alter the kinetics of protein denaturation at the AWI that has led to the near-atomic resolution (3.3–4.6 Å) cryoEM structures of a relatively small protein (117 kDa human PreP) that exhibits high conformational heterogeneity. Furthermore, PreP provides an intriguing model protein to assess the efficacy of newly emerging theories and practices of grid chemistry and vitrification process, e.g., VitroJet or Shake-it-off, aimed at preventing protein denaturation during vitrification[48,49].

Our integrative structural approaches lead to the formulation of the following model for the catalytic cycle of PreP (Fig. 5A).

**Table 2 Data collection and structure refinement statistics of MB60-bound PreP.**

| Data collection | |
|---|---|
| Beamline | APS-19ID |
| Wavelength (Å) | 0.9792 |
| Space group | C2 |
| Cell dimension(Å) | |
| $\quad$ a, b, c | 245.6, 85.5,158.2 |
| $\quad$ $\alpha$, $\beta$, $\gamma$ | 90.0, 127.5, 90 |
| Resolution (Å) | 44.85–2.27 (2.31–2.27) |
| $R_{meas}$ (%)[a] | 18.6 (80.1)[e] |
| $R_{p.i.m}$ (%)[b] | 9.7 (42.8)[e] |
| $CC_{1/2}$[c] | (0.639)[e] |
| CC*[d] | (0.883)[e] |
| I/sigma | 18.2 (2.1)[e] |
| Redundancy[f] | 3.5 (3.3)[e] |
| Completeness (%) | 99.9 (98.0)[e] |
| Unique reflections | 119317 |

| Refinement | |
|---|---|
| $R_{work}$[g] | 0.176 |
| $R_{free}$[h] | 0.208 |
| No. of atoms | |
| $\quad$ Protein | 15,861 |
| $\quad$ Water | 811 |
| B-factors | |
| $\quad$ Protein | 36.1 |
| $\quad$ Substrate | 33.5 |
| $\quad$ Water | 40.5 |
| R.m.s. deviations | |
| Bond lengths (Å) | 0.006 |
| Bond angles (°) | 0.972 |
| Ramachandran plot (%) | |
| $\quad$ Favorable region | 92.6 |
| $\quad$ Allowed region | 7.4 |
| $\quad$ Disallowed region | 0 |
| $\quad$ PDB code | 4RPU |

[a]$R_{meas} = \Sigma_{hkl} \left[ n/(n-1) \right]^{1/2} \Sigma_i \left| I_{hkl,i} - \langle I_{hkl} \rangle \right| / \Sigma_{hkl} \langle I_{hkl} \rangle$
[b]$R_{p.i.m.} = \Sigma_{hkl} \left[ 1/(n-1) \right]^{1/2} \Sigma_i \left| I_{hkl,i} - \langle I_{hkl} \rangle \right| / \Sigma_{hkl} \langle I_{hkl} \rangle$
[c]$CC_{1/2}$—Pearson correlation coefficient between random half-datasets—$\rho_{x,y} = \text{cov}[(x,y)/(\sigma_x \sigma_y)]$
[d]$CC^* = [2CC_{1/2}/(1 + CC_{1/2})]^{1/2}$
[e]The outer resolution shell. Values in parentheses indicate the highest resolution shell.
[f]$N_{obs}/N_{unique}$.
[g]$R_{work} = \Sigma_{hkl} \left| |F_{obs}| - k | F_{calc} | \right| / \Sigma_{hkl} | F_{obs} |$.
[h]$R_{free}$, calculated the same as for $R_{work}$ but on the 5% data excluded from the refinement calculation.

Both cryoEM and SAXS data indicate that apo-PreP prefers to be in the partially open state, which cannot capture substrates. The C-terminal end of the switch B helix (e.g., Met 436) is located at the region where PreP-C pivots away from PreP-N during the closed to open transition (Figs. 3d and 5A). The rigid body motion of PreP-N and PreP-C entropically drives the separation between PreP-N and PreP-C domains. Governed by the rotation of the switch B region, the rigid body motion between PreP-N and PreP-C can trigger the conformational switch of the extended loop in the switch C region, allowing PreP to transition into the open state (Figs. 3c–e and 5A). Presequences rich in positively charged residues can then be attracted to the negatively charged surface of the PreP-N catalytic chamber (Figs. 2f and 5A). Furthermore, the high dipole moment of Aβ permits the charge complementation of this peptide with the catalytic chamber formed by PreP-N and PreP-C, which are negatively and positively charged, respectively (Figs. 2f and 5A). Plentiful hydrophobic residues in presequences and Aβ, or the small molecule inhibitor MB60, then interacts with the hydrophobic exosite to promote the rotation of switch B helix that induces the open to closed transition (Figs. 3f and 5A). The hydrophobic residues of PreP substrates also interact with the hydrophobic pocket at the catalytic site formed by PreP-N and PreP-C, further promoting the favorable interaction between PreP-N and PreP-C (Figs. 3f and 5A). The motion between PreP open and closed states, in conjunction with the selective interaction between the PreP catalytic chamber and the peptide substrate, provide the requisite force to unfold presequences and Aβ. This leads to the exposure of a β-strand within the presequence peptide or Aβ that can complement and stabilize the catalytic cleft formed in part by the switch A region, which in turn facilitates proteolysis. PreP then transitions from the closed state to the open state to release the reaction products. The transition between PreP open and partially open/closed state occurs quite frequently because its rate needs to be faster than the rate of Aβ degradation, which is 50–200 per second[6]. The model described above provides the molecular basis for the key conformational changes during the PreP catalytic cycle that facilitate amyloidogenic peptide capture and degradation. As the loss of function mutations in human PreP are associated with neurological disorders, e.g., cognitive impairments/disability and cerebellar atrophy[10,11], our model should provide guidance for future investigation into how to boost PreP activity for better control of mitochondrial proteostasis.

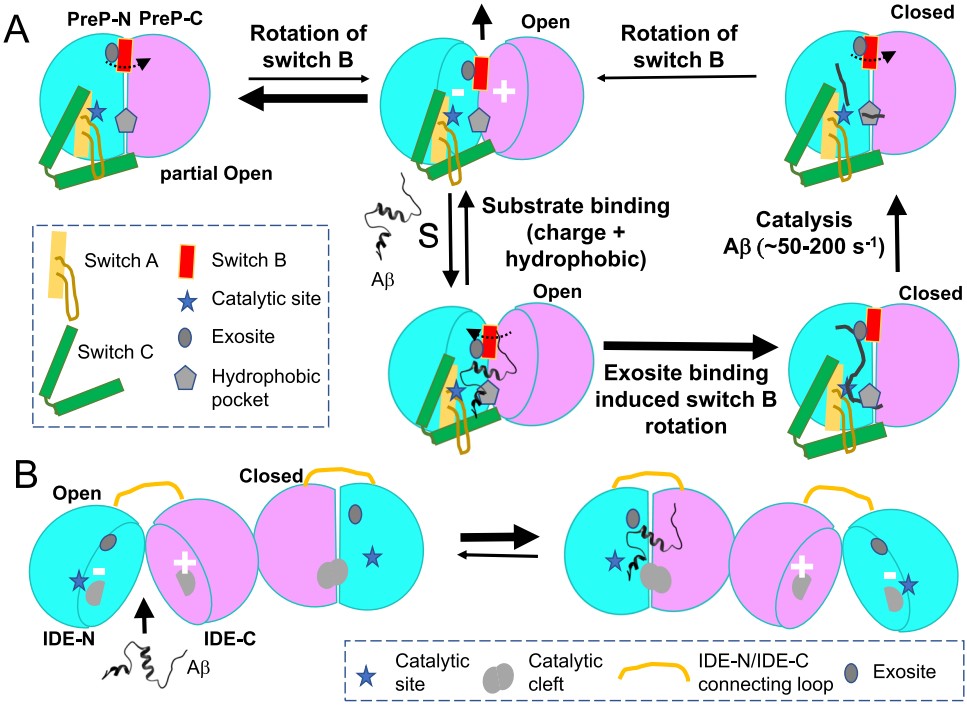

**Fig. 5 Comparison of key structural features of PreP and IDE for catalysis. A** A model of the catalytic cycle of PreP to depict the structural basis for the conformational switch and substrate recognition. PreP-N, PreP-C, switch A, switch B, and switch C domains are colored in cyan, magenta, yellow, red, and green, respectively and the color scheme is used throughout figures. **B** A model of dimeric IDE to depict the allosteric regulation. The domains and switch regions are depicted as the cartoon and colored the same as the figures above. A detailed description is in the discussion.

Our structural analysis of two members of the cryptidase family, IDE, and PreP, indicates that they have distinct mechanisms of substrate suggests a common framework for amyloidogenic peptide recognition and regulation (this work) with distinct specializations that support efficient cytotoxic peptide clearance in their distinct cellular niches[6,23,37,50,51]. Both enzymes belong to the M16 clan of metalloproteases and have homologous ~55 kDa N- and C-terminal domains (Fig. 5)[23,51]. Both enzymes also undergo the open-closed transition during their catalytic cycle and only the open state can capture substrate and release proteolytic products (Fig. 5)[6,23,51]. Furthermore, their catalytic cleft of both enzymes is formed between their N- and C-terminal domains, which is unstable in the absence of substrate. They use a substrate-assisted catalysis mechanism to selectively recognize and degrade amyloid peptides (this work)[37,50]. However, there are noticeable differences between IDE and PreP. Firstly, how these two enzymes open up is quite different due to the connecting region between their N- and C-terminal domains (Fig. 5). PreP is connected by the relatively long, dual α-helical hairpin switch C region that is connected to the zinc-containing D1 domain via the switch A region. Consequently, PreP undergoes a "book-opening" motion along a rather large surface between the PreP-N and PreP-C domains that is guided by the rotation of the switch B region. However, the short ~40 amino acid long loop of IDE allows IDE-N and IDE-C domains to pivot along a much smaller surface between D2 and D3 domains for a "packman-like" open-close motion, leading to the maximal separation between D1 and D4 domains[37]. The second difference is their oligomerization state (Fig. 5). The monomeric PreP 'rests' in a partially open state and the transition between partial open and open state allows PreP to capture its substrates. Contrary to monomeric PreP, IDE exists as a dimer to allow for the allosteric regulation of catalysis[37,50]. Specifically, at

least one of the two subunits within the IDE dimer is typically in the open state and primed for substrate capture most of the time. The binding of substrate allosterically facilitates the opening of the other IDE subunit within the IDE dimer, leading to enhanced IDE catalysis[37,50]. Furthermore, these two enzymes use the exosite located at their D2 domain to recognize different structural features of their substrates (Fig. 5). The PreP exosite recognizes hydrophobic residues of peptide substrates while the IDE exosite binds the N-terminal mainchains[6,23]. Despite these profound differences, we have shown that the structural analyses can be used to rationally design mutations to enhance the enzymatic activities of both enzymes (this work)[50]. Furthermore, small molecules that can either boost IDE activity or selectively inhibit the degradation of insulin by IDE has been discovered[52-54]. Future work will realize the preventative or therapeutic potential in controlling the proteolytic activity of these Aβ-degrading proteases for neurodegenerative diseases caused by amyloid peptide-mediated toxicity.

## Methods

**Expression and purification of PreP**. The expression vectors for wild-type human PreP and E107Q mutant were made previously as described[6]. Vectors for other PreP mutants were made using QuikChange site-directed mutagenesis kit with the following primers: PreP N358A-5′cccGCttctcccttttacaaagccttg3′ & 5′gggagaaGCgggcccagaagtcaaga3′, PreP E367A-5′ttgattgCatctggccttggcacagactttc3′ & 5′gccagatGcaatcaaggctttgtaaaagggag3′, PreP P558G-5′tgtctgGcagcgttgaaagtttccgatattg3′ & 5′cgctgCcagacaagaggcatcttgagg3′, PreP N593D-5′accGatggcatggtgtatttccggg3′ & 5′gccatCggtgggctgggcgcagtactg3′, PreP Q637A-5′caggctGCgcagatagaattgaagaccggagg3′ & 5′tatctgcGCagcctgctcccggtagtcaag3′, PreP E901V-5′attcgagTaaaaggcggtgcttatggtgg3′ and 5′gcctttActcgaatttctgtatgcaagaatttgg3′. Wild-type human PreP and various mutants were expressed and purified as described[6]. Briefly, *E. coli* Rosetta(DE3) containing the expression plasmid for the desired PreP construct was grown in T7 medium at 25 °C with 300 μM IPTG induction for 20 h. Cells were harvested via centrifugation at 10,000×*g* for 20 min (4 °C) and the resulting pellet was resuspended in a solution containing 20 mM Tris pH 8, 500 mM NaCl, 0.5 mM EDTA (omitted for purification of active PreP) 0.3 mM phenylmethylsulfonyl fluoride (PMSF), and 1 mM benzamidine-

HCl. Proteins were then purified over a $Ni^{2+}$-NTA affinity column equilibrated with a solution containing 20 mM Tris pH 7.7, 100 mM NaCl, and 0.1 mM PMSF. Bound protein was washed to baseline on-column first with a solution containing 20 mM Tris pH 7.7, 0.1 mM PMSF, and 500 mM NaCl followed by that containing 20 mM Tris pH 7.7 0.1 mM PMSF 50 mM NaCl, and 5 mM imidazole, to remove weakly bound contaminants. Protein was then eluted into a solution containing 20 mM Tris pH 7.7, 0.1 mM PMSF, 50 mM NaCl, and 150 mM imidazole. The protein sample was then diluted to a NaCl concentration <25 mM and loaded on a Source Q anion exchange column equilibrated with a solution containing 20 mM Tris pH 8.0 and 0.1 mM PMSF. After loading, the column was washed to baseline with the same equilibration buffer and bound protein was eluted with a solution containing 20 mM Tris pH 8.0, 0.1 mM PMSF, and a linear gradient of NaCl from 0 to 1 M over 25 column volumes. The resulting protein peak containing PreP was then applied to a Superdex 200 column equilibrated with a solution containing 20 mM Tris pH 8.0 and 50 mM NaCl for SEC. Protein purity was assessed via SDS-PAGE, aliquots were flash-frozen in liquid nitrogen and stored at −80 °C.

**Differential scanning fluorimetry**. To optimize conditions for cryo-EM data collection, DSF was applied to screen 40 buffers and 98 additive conditions. The DSF was carried on with about 1 mg/ml PreP and 10× Sypro Orange in 20 µl buffers using Thermo Fisher Step ONE RT-PCR. Using the melting temperature and slope as selection criteria, the following condition was identified as best for the grid making with 20 mM Tris (pH 7.7), 150 mM NaCl, 10 mM EDTA, 0.5 mM β-mercaptoethanol.

**CryoEM data collection and analysis**. Purified PreP was further purified by Superdex 200 chromatography using a buffer containing 20 mM Tris, pH 7.7, 150 mM NaCl, 10 mM KCl, 20 mM EDTA, and 1 mM β-mercaptoethanol. Grids were prepared using either Vitrobot or chameleon[35]. For Vitrobot grids, Quantifoil holey carbon-coated 200 mesh copper R1.2/1.3 grids were plasma cleaned in the air for 30 s using a Solarus plasma cleaner (Gatan). Totally, 2.5 µl of the sample was applied to the grid, waited for 15 s, and blotted with one layer of standard vitrobot filter paper (grade 595 with the outer/inner diameters of 55/20 mm, respectively; Ted Pella, inc. #47000-100), force 1, 100% humidity, for 2.5–3.5 s from both sides of the grids followed immediately by plunging into liquid ethane. For chameleon prepared grids, 300 mesh carbon or gold lacey nanowire grids were plasma cleaned with $O_2$ and $H_2$ for 10 s using a Solarus plasma cleaner (Gatan). The grids were plunged at 133 ms. All images were acquired using a Titan Krios microscope (FEI) operated at 300 keV with a Gatan K2 direct electron detector (Gatan) in counting mode. Images were automatically acquired using Leginon[55] using collection parameters as shown in Table 1. Images were processed using software integrated into RELION3.0[56]. Frames were aligned using MotionCor2 software with electron exposure weighting[57], CTF was estimated using Gctf[58], particles were picked and extracted automatically using RELION3.0[56]. Particle stacks were processed through several rounds of 2D and 3D classification. Example images and 2D class averages are shown in Supplementary Figs. 3, 7, 9, and 10. Selected classes with good sphericity values (ranging from 0.837 to 0.878) based on 3D-FSC[59] were then processed for high-resolution 3D refinement (Table 1 and Supplementary Figs. 3, 7, 9, and 10). Finally, the overall map was improved by particle polishing in RELION3.0[56]. The final resolution was estimated using gold-standard Fourier Shell Correlation (FSC = 0.143) (Supplementary Figs. 4, 8, 9, and 10). CryoEM data collection and processing statistics are listed in Table 1. Structural models were built using Aβ-bound PreP crystal structure (PDB = 4NGE) as a template[6]. Density fitting and structure refinement were performed using UCSF CHIMERA[60], COOT[61], and PHENIX[62]. The refinement statistics are listed in Table 1.

**CryoET analysis**. Tilt-series were collected with Leginon[55] on a Titan Krios with a Gatan K2 counting camera using the same sample preparation as for cryoEM. Tilt-series were aligned with Appion-Protomo[63], electron exposure-weighted using equation 3 in Grant and Grigorieff[64], and reconstructed with Tomo3D[65]. Sub-tomogram processing, including particle picking, alignment, and classification, was performed with Dynamo[66] and reconstructed with Tomo3D[65].

**Protein crystallization, data collection, and structure determination**. PreP E107Q was modified by reductive lysine methylation prior to Superdex 200 chromatography as reported previously[6]. Specifically, 1–10 mg/ml PreP in buffer containing 50 mM HEPES pH 8.0, 500 mM NaCl, 5% Glycerol (v/v), and 10 mM β-mercaptoethanol was incubated with 40 mM formaldehyde and 20 mM dimethylamine-borane complex for 2 h at 4 °C, followed by an overnight incubation with an additional 20 mM dimethalymine-borane complex. The reaction was then quenched for 2 h by adding glycine to 13.3 mM and DTT to 5 mM. Totally, 5–7 mg/ml lysine-methylated PreP-E107Q in buffer containing 20 mM HEPES pH 7.5, 250 mM NaCl, 2 mM DTT, and 200 µM MB60 (MolPort #001-620-747) was combined with mother liquor containing 15.0% (w/v) PEG 8000, 15 mM TCEP, 80 mM sodium cacodylate pH 6.7, 160 mM calcium acetate, and 20% (v/v) glycerol in a 1:1 (v/v) ratio for the crystallization of PreP-MB60 complex by hanging-drop vapor diffusion at 18 °C. Crystals grew for one week prior to data collection. Crystals were cryoprotected in mother liquor containing 30% (v/v) glycerol, then flash frozen in liquid nitrogen. Diffraction data were collected at beamline 19ID at

Argonne National Laboratory and processed using HKL3000[67]. The structure of PreP in complex with MB60 was determined by molecular replacement using Phaser and PreP structure (4L3T) as the search model. Model building—including the addition of missing 317–323 residues in chain A—were performed using COOT[61] and refinement was done using PHENIX[62]. The final 2.27 Å resolution model (pdb = 4RPU) has $R_{work}$ = 18.6% and $R_{free}$ = 20.6%. Data collection and structure refinement statistics are listed in Table 2. Presumably, due to the shorter time of crystallization, only cysteine 112 was modified with the dimethylarsenic moiety while cysteine 556 was not. Cysteine 556 is in close proximity with cysteine 119 to form a disulfide bond. The absence of a disulfide bond between these residues might be due to the presence of a reducing agent during the purification and/or crystallization.

**SAXS data collection and analysis**. SAXS data were collected at the BioCAT/18ID beamline at Advanced Photon Source, Argonne National Laboratory (Chicago, USA) beamline 12ID-B, at 23 °C using 1.1 mg/ml protein and an incident X-ray wavelength of 0.886 Å, and protein concentration of 0.5 mg/ml. For MB60-binding experiments, PreP was preincubated with 200 µM MB60 on ice prior to SAXS data collection. Data were reduced and analyzed using ATSAS[68] using the photon counting PILATUS 3 1 M at room temperature (23 °C) and an incident X-ray wavelength of 1.03 Å. The 3.5 m sample-to-detector distance yielded a range of 0.005–0.33 Å$^{-1}$ for the momentum transfer ($q = 4\pi \sin\theta/\lambda$ where $2\theta$ is the scattered angle between the incident and scattered beam and λ the X-ray wavelength). The PreP sample was loaded onto an SEC system (ÄKTA pure, GE Healthcare Life Sciences, Piscataway, NJ) with a GE Superdex 200 10/300 G. Totally, 2–3 mg protein was injected to Superdex 200 in the buffer containing 20 mM Tris, pH 7.7, 100 mM NaCl with/without EDTA. To remove the zinc ion from PreP, the protein was dialyzed against 500 ml 20 mM Tris pH7.7, 100 mM NaCl, 20 mM EDTA. A 5-fold molar excess of Aβ, CS27, or MB60 was mixed with PreP immediately prior to loading on the Superdex 200 in the buffer containing 20 mM Tris-HCl (pH 8.0), 100 mM NaCl, with/without 20 mM EDTA. The data were reduced and analyzed using ATSAS[68]. PRIMUS and GNOM in the ATSAS suite were used to determine the $R_g$ value in reciprocal and real space, respectively[68]. $D_{max}$ and $P(r)$ distribution were calculated by GNOM. Disordered regions were modeled using the structures based on the Alphafold structure of PreP (AF-Q5JRX3-F1)[68] and theoretical scattering curves for different models were generated and fit to the experimental data using CRYSOL in the ATSAS suite[68]. The addition of missing segments into the experimental structures has been shown to substantially improve the fitting of SAXS data[69]. EOM was performed using EOM 2.0[38–40]. With the N- and C-domains of PreP as input, 10,000 native-like chain models with 200 points were generated and assessed over 100 cycles of a genetic algorithm using default parameters. OLIGOMER in the ATSAS suite was used to determine the percent composition by parsimonious conformational states that best fit the observed data[68]. Key parameters in SAXS data acquisition, sample details, data analysis, and modeling fitting, as well as the software used, are listed in Supplementary Table 2 as recommended by publication guideline[70].

**Hydrogen deuterium exchange–mass spectrometry**. Prior to performing comparative H/D exchange experiments, enzymatic and quench conditions that produced an optimal fragmentation pattern of PreP were established as previously described[71,71]. Briefly, for PreP and MB60 study, 3 µl 4.7 mg/ml PreP in buffer containing 20 mM Tris-HCl (pH 7.5), and 50 mM NaCl was diluted with 9 µl of buffer A (8.3 mM Tris-HCl (pH 7.5), 50 mM NaCl in $H_2O$) at 0 °C. For PreP and substrates study, 3 µl 10.5 mg/ml PreP in buffer containing 20 mM Tris-HCl (pH 7.7), 150 mM NaCl and 10 mM EDTA was diluted with 9 µl of buffer B (8.3 mM Tris-HCl (pH 7.7), 150 mM NaCl, 10 mM EDTA in $H_2O$) at 0 °C. The sample was then mixed with 18 µl of ice-cold quench buffers containing 0.8% formic acid, 16.6% glycerol, and various concentrations of GuHCl (0.08, 0.8, and 1.6 M). Quenched samples were then subjected to HDX-MS apparatus for proteolysis and LC/MS analysis. The use of 0.8 M GuHCl resulted in the best sequence coverage of PreP. For HDX-MS analysis, 13 µM PreP in the presence or absence of 130 µM MB60 with buffer 8.3 mM Tris-HCl pH7.2, 50 mM NaCl, and 2.1% DMSO in $H_2O$, 260 µM Aβ, or 260 µM CS27 in buffer B and 2.1%DMSO in $H_2O$ was incubated at room temperature for 30 min prior to chilling to 0 °C for deuteration studies. Functional hydrogen-deuterium exchange reactions were initiated by adding a 3 µl sample into 9 µl of buffer A or buffer B in $D_2O$ (pD$_{READ}$ = 7.2) and incubated at 0 °C for 10, 100, 1000, 10,000, and 100,000 s[72]. The exchange reaction was terminated by adding 18 µl of ice-cold 0.8% formic acid, 0.8 M GuHCl, 16.6% glycerol for a final pH of 2.5. Quenched samples were then immediately frozen on dry ice and stored at −80 °C prior to LC/MS analysis. Un-deuterated and equilibrium-deuterated control samples are also prepared as previously described[73]. Frozen samples were later loaded onto a cryogenic autosampler[74], thawed at 4 °C, then passed over an immobilized pepsin column (16 µl bed volume) for 30–40 s digestion. Proteolytic fragments were collected on a trap column and separated using Optimize Technologies C18 reverse-phase analytical column (Halo EC-C18 0.2 × 50 mm, 2.7 µm) with an acetonitrile linear-gradient (6.4–38.4% over 30 min). The effluent was directed into an OrbiTrap Elite Mass Spectrometer (Thermo-Fisher Scientific, San Jose, CA). Instrument settings were optimized to minimize the back-exchange[75]. The data was acquired in either MS1 profile mode or data-dependent MS/MS mode. Peptide identification was done with the aid of Proteome

Discoverer software (ThermoFisher). Mass envelope centroids of deuterated peptides were calculated with HDEXaminer v2.5.1 (Sierra Analytics Inc, Modesto, CA) then converted to corresponding deuterium incorporation with corrections for back-exchange[76]. Deuterium uptake plots, Heat Map and Difference Maps were generated with Excel macro and MatLab scripts. Key parameters and data are included in Supplementary Table 3 and Data 1–2 according to the recommendation[77].

**Enzymatic assay**. The fluorogenic peptide substrate Mca-Y-V-A-D-A-P-K(Dnp)-OH (R&D Systems, Catalog # ES007) was used to measure the activity of PreP. The reaction was monitored on a Synergy Neo microplate reader using an excitation wavelength of 320 nm and an emission wavelength of 405 nm. Reactions were carried out at 37 °C, using 5 nM PreP with various concentrations of substrate V (5, 10, 20, or 40 μM) in 200 μL of buffer containing 20 mM Tris, pH7.7, 150 mM NaCl, 1 mM β-mercaptoethanol. Degradation of substrate V was assessed by monitoring the fluorescence increase for 10 min at 30-s intervals. To calculate enzymatic activity, background subtraction and linear regression fitting were used to find the initial velocity, whereas specific activity was determined by comparing the maximal fluorescence converted from the known quantity of substrate V by PreP.

**Cellular proliferation assay of human K562 leukemia cells**. MB60 was purchased from MolPort. Leukemia K562 cells that express GFP (gifted from Luke Gilbert; engineered using K562 cells from ATCC CCL-243) were grown in RPMI-1640 medium with 10% FBS, 0.05 mM 2-Mercaptoethanol, and penicillin/streptomycin. Cells at $1 \times 10^5$ viable cells/mL were added to a 96-well plate with 100 μl/well. MB60 was added to the indicated concentrations, 0.8 to 25 μM. Cell growth was monitored continually every 4 h up to 60 h using IncuCyte S3s (Essen BioScience). The cell count from 12 to 40 h was used to calculate their cell doubling time.

**Reporting summary**. Further information on research design is available in the Nature Research Reporting Summary linked to this article.

## Data availability

The data supporting the findings of this study are available from the corresponding author upon reasonable request. Structure factor amplitudes and coordinates for the crystal structures of MB-60 bound PreP and Aβ-bound PreP are deposited in the Protein Data Bank under accession number 4RPU and 4NGE, respectively. The 3D cryoEM density maps generated in this study is deposited in the Electron Microscopy Data Bank under accession codes EMD-22278 (Apo-PreP pC1 state), EMD-22279 (Apo-PreP pC2 state), EMD-22280 (Apo-PreP open state), EMD-22281 (Aβ-bound PreP), and EMD-22282 (CS27-bound PreP). The corresponding atomic coordinates are deposited in the Protein Data Bank under accession numbers 6XOS (Apo-PreP pC1 state), 6XOT (Apo-PreP pC2 state), 6XOU (Apo-PreP open state), 6XOV (Aβ-bound PreP), and 6XOW (CS27-bound PreP). EM data in the form of unprocessed micrographs is deposited in the Electron Microscopy Public Image Archive (EMPIAR) under accession number EMPIAR-10937. The tomogram shown in the figures is deposited to the EMD with the accession number EMD-25921. Raw tomography data is deposited to EMPIAR with the accession number EMPIAR-10929. The HDX-MS data are deposited in ProteomeXchange under the accession number PXD029542. SAXS data is deposited in the Small Angle Scattering Biological Data Bank (SASBDB) under accession codes SASDKK3 (Apo-PreP), SASDKL3 (MB60-PreP), SASDKM3 (CS27-PreP), and SASDKN3 (Aβ-PreP). Source data are provided with this paper.

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

## Acknowledgements

We are grateful to Luke Gilbert for the leukemia K562 cell line labeled with GFP and Srinivas Chakravarthy at BioCAT, APS for assisting with SAXS data collection and analysis. This work was supported by the NIH grant GM121964 to W.-J. Tang, GM103622 to Tom Irving at BioCAT, APS. Some of this work was performed at the Simons Electron Microscopy Center and National Resource for Automated Molecular Microscopy and National Center for CryoEM Access and Technology located at the New York Structural Biology Center, supported by grants from the Simons Foundation (SF349247) and the NIH National Institute of General Medical Sciences (GM103310, U24 GM129539). Use of the Advanced Photon Source was supported by the U.S. Department of Energy, Office of Basic Energy Sciences, under contract No. DE-AC02-06CH11357.

## Author contributions

W.-J.T., W.G.L., S.L., B.C., C.S.P., and M.Z. designed the project. W.G.L. performed cryoEM grid preparation, data acquisition, and processing assisted by H.W., M.P., and C.L. and overseen by W.-J.T., M.Z., B.C., and C.S.P. A.J.N. performed cryoET data acquisition, and A.J.N., W.G.L., and J.M.M. performed the analysis. W.G.L., W.-J.T., and M.Z. built and refined cryoEM structural models. W.G.L. and S.M. performed protein purification and crystallographic data collection, built, and refined structural models. W.G.L. performed protein purification for HDX-MS and D.L. and S.L. performed HDX-MS and analysis. W.G.L. purified proteins and W.G.L. and J.M.M. performed SAXS studies. J.W. performed the inhibitor screen and characterization of MB60 and J.W. and C.M.K. provided critical reagents for the crystallographic analysis. W.G.L., W.-J.T., J.V.L.K., J.M.M., and S.L. wrote the paper. S.M., J.W., M.Z., and B.C. contributed to the paper finalization.

## Competing interests

The authors declare no competing interests.
