## [Peer Review File · Nature Communications]

REVIEWER COMMENTS

Reviewer #1 (Remarks to the Author):

The article does seem to provide a comprehensive cryo EM, and biophysical analysis of the the various conformation States of the peptidase, Prep. I am no expert in cryoEM, but the presentation and results seem to support the conclusions drawn and are in line with what I would expect to see presented. It is interesting to see the utility of cryoEM to define the conformational states of a mixed system.

I am a SAXS expert, and I do feel that the SAXS analysis is underdone. While it is nice to see a quantification of a mixed population, using form-factors generated from the cryoEM structures, there is no presentation of the quality of the fits for the OLIGOMER analysis. There needs to be a presentation of the Guinier plots, to provide confidence in the radius of gyration values presented. There's little discernible difference in the profiles presented in the Guinier region, so a proper presentation of that analysis is required to assess whether the R_g values are valid. I would like for there to be further analysis using ensemble optimisation modelling, as if there are differences in the open and closed states there should be observable differences in the distribution of structures that can be generated via this method, and thus it would support the results indicating a distribution of conformational states are present. I strongly suggest that the authors consult "Trehwella et al, 2017, 2017 publication guidelines for structural modelling of small-angle scattering data from biomolecules in solution: an update" for what is recommended to be presented for the publication of SAXS data. At the very least, fit quality should be presented in some fashion.

Otherwise the results seem fairly comprehensive, and the general conclusion that there are multiple conformational states present seems supported.

Minor comments – there seems to be some poor sentence structure, and misuse of tense throughout the manuscript. I suggest a through grammar and sentence clarity check.

Reviewer #2 (Remarks to the Author):

In the article "Structural basis for the mechanism of human prosequence protease conformational switch and substrate recognition", the authors utilized advanced vitrification process, the combination of self-blotting nanowire grids and piezo dispensing, during sample preparation for CryoEM and successfully obtained high-resolution structural information for three open-states

Presequence protease (PreP) together with two bound-states PreP. With further integration of size-exclusion chromatography (SEC) - small angle X-ray scattering (SAXS) and hydrogen deuterium exchange (HDX) – mass spectrometry (MS), this study proposed a model for the catalytic cycle of PreP, providing structural basis for the transition between different states of PreP as well as the selectivity and capture principle of its substrates. In addition, the comparison with the catalytic mechanism of insulin degrading enzyme (IDE) suggest a typical framework for the amyloidogenic peptide catalysis, paving the road for further tailoring the enzymatic activities. In general, this study fits the interests of the broad audience of Nature Communications, recommended for publication after addressing the comments below.

Comments:

1. In the last paragraph of the “Mechanism for the conformational switches between PreP open and closed states in the presence and absence of substrate”, the description is confusing. The expectation before HDX analysis is not unclear. Specifically, the substrate binding sites nor the binding interfaces between PreP-C and PreP-N were not mentioned in terms of residues (e.g., aa xx-xx). However, HDX protection observed in region aa 703-720, aa 892-940 are concluded as binding interfaces that meets author’s expectation. Please justify the logic and label the regions on corresponding figures.
2. Among these protected regions, aa 91-121 is not labeled nor discussed in the manuscript. Please justify the decrease of deuterium uptake upon substrate binding.
3. In Figure S17, the apo state of peptide 126-133 experiences a decrease in deuterium uptake at the final time point. Please explain the possible reasons.
4. Substrate binding shows different structural impact on Switch A, B and C domains, among which Switch B shows a less extent HDX protection. How does this observation relate to the proposed working mechanism? Please describe.
5. This paper needs better organization, especially for Figure S15 and S16. MB-60 is discussed after substrate binding, however, is shown in front.
6. Make sure the correct figures are referred in each paragraph. For example, Figure S14-16 are wrongly assigned, should be 15-17. Figure S14-15 is also wrong, should be 15-18. Please correct and check carefully throughout.
7. In the case of MB-60 binding, aa 114 – 144 experienced considerable amounts of HDX protection, however, is not discussed. Another missing peptide is aa 630-635. Please justify the observation and how can that relate to the inhibition process.
8. Please report key parameters for HDX analysis. For example, any smoothing functions to get the ΔD plots between different states? What is the value?
9. In the introduction, there are missing spaces in (HDX) – mass spectrometry. Please correct.
10. Please keep the terminology consistent throughout the manuscript, HDX or DXMS.

11. The label "Figure 1" is missing in Figure 1. Please correct.
12. Please keep consistency on the color code of the same domain and label. In Figure 2, the protein "Switch A" should be yellow.
13. In Movie 5, the Switch B is mislabeled. Please correct.

Reviewer #3 (Remarks to the Author):

In this manuscript the authors describe a collection of structure of apo-PreP open states along with A β - and citrate synthase bound PreP at 3.3 Å-4.6 Å resolution. The authors use a combination of cryo-EM, SEC_MALS and HDMX to investigate the details of PreP catalytic cycle and conformational changes. This paper is intended to a broad audience given the importance of the subject and technical applicability of the methods used to overcome the limitations due to the small molecular weight of the complex and preferential view.

General Comment:

Given the lack of line indicators in the manuscript it is very challenging to indicate where to make edits.

Points to address:

1- The authors indicate that the use of the Spotiton dramatically improve sample preparation, particle orientation and air water interface but they lack to show tomograms for these samples. I think this panel would be extremely useful.

2- Can the authors comment on the unusual behavior of the FSC curves presented in Figure S8C? This is generally a behavior observed for datasets still presenting a large amount of heterogeneity. I wonder if subclassification was attempted.

3- The EM data acquisition and processing have been performed expertly, but a few minor details should be addressed.

Most importantly, both samples appear to have a rather strong preferred orientation in ice. Certain regions of Euler space are overpopulated, and thus the resolution of the reconstruction is not isotropic. It is likely that both the global and local resolutions of the structures based on the FSC are inflated due to this preferred orientation. This should be addressed by reporting the anisotropy of the reconstruction by one of several reported methodologies (from the labs of Chris Russo, Dmitry Lyumkis), and the data should be reprocessed with normalization of Eulers to reduce the impact of the preferred views

on the resolution.

4- Given the importance of the vitrification for obtaining a sample amenable for structure determination, details regarding blotting paper used.

5- Since "dose" is a volumetric measurement and reported in A3, "electron exposure" (or fluence) and "exposure rate" (or flux) should be used in the methods and Table S1.

Minor point:

1- There is a repetition in the first section of the results: "Apo-PreP grids were then prepared by Vitrobot and a dataset of 2,626 micrographs collected at 300 kV on a Titan Krios."

2- Following this section there is another one for which the meaning is not necessarily clear: "The two major classes, comprising ~208,000 and ~118,000 particles, displayed an intact PreP-N domain but a denatured PreP-C domain and were refined to 4.2 Å (Fig. 1B, Extended data Fig. 4,5)". Is "but" here placed instead than "and"?

3- In figure S5 the last two helices and structure are different. Could the rendering be made in the same way to have self-consistency within the figure?

Responses to REVIEWER COMMENTS (highlighted in bold and italic)

Overall response:

We first like to thank all three reviewers for your careful review and your comments have allowed us to substantially improve our manuscript. We have done substantial additional data acquisition and analysis for single particle EM, SAXS, and HDX-MS. Specifically, we have repeated the HDX-MS analysis of PreP in the presence or absence of amyloid beta and MB60 to fulfill the required n=2 analysis. We also re-analyzed SEC-SAXS data using both oligomer and ensemble optimization modeling as required. We have used 3D-Fourier Shell Correlation (FSC) algorithm described in Tan et al Nature Methods 2017 to show the acceptable orientation bias of our cryoEM data. The requisite details in data acquisition and analysis of SAXS and HDX-MS are included in new tables as recommended by Trehwella et al 2017 and Masson et al 2019, respectively. Furthermore, the HDX-MS data is deposited in the requested site and included in the tables in extended data. The specific responses to reviewers' comments are as below.

Reviewer #1 (Remarks to the Author):

The article does seem to provide a comprehensive cryo EM, and biophysical analysis of the the various conformation States of the peptidase, Prep. I am no expert in cryoEM, but the presentation and results seem to support the conclusions drawn and are in line with what I would expect to see presented. It is interesting to see the utility of cryoEM to define the conformational states of a mixed system.

I am a SAXS expert, and I do feel that the SAXS analysis is underdone. While it is nice to see a quantification of a mixed population, using form-factors generated from the cryoEM structures, there is no presentation of the quality of the fits for the OLIGOMER analysis. There needs to be a presentation of the Guinier plots, to provide confidence in the radius of gyration values presented.

Guinier plots are included in Extended data Figure 13.

There's little discernible difference in the profiles presented in the Guinier region, so a proper presentation of that analysis is required to assess whether the Rg values are valid. I would like for there to be further analysis using ensemble optimisation modelling, as if there are differences in the open and closed states there should be observable differences in the distribution of structures that can be generated via this method, and thus it would support the results indicating a distribution of conformational states are present.

The structural models we used to originally calculate the scattering profiles for the pC, pO, and O states were missing ~50 residues in a loop region and at the N-terminus. We have found that such ~5% missing segments can profoundly reduce the quality of fit for the experimentally determined structural models to the SAXS experimental data (Liang & Mancl et al Structure 29:709-720, 2021). Thus, we modeled in the missing residues based on the AlphaFold structure of PreP (AF-Q5JRX3-F1), recalculated the scattering profiles, and updated the necessary figures. We have included Guinier plots with R² values and data/fit plots with χ^2 values in extended data figure 13 for the experimental data and scattering profiles calculated from our structural models. We have re-run the OLIGOMER analysis with the full-length models and the results are consistent with the previous analysis. Guinier and data/fit plots for the OLIGOMER analysis with statistics are included in extended data figure 13. We performed ensemble optimization modeling (EOM) using EOM 2.0. For this analysis, the N- and C- domains of the full-length PreP models were provided as two input domains.

From this input, 10,000 native-like models were generated. Scattering profiles were assessed over 200 points using the default parameters prior to running the genetic algorithm. The results have been added to figure 2 and extended data figure 13 along with Guinier, data/fit plots, and ensemble composition with relevant statistics. EOM produced two or three model mixtures for each of the four experimental conditions. For all conditions, the major classes resulting from EOM showed high similarity to the states observed in our cryoEM analysis. However, several conditions produced minor classes that appeared to be physiologically irrelevant and opened to a degree beyond that which has been previously observed experimentally. These minor classes inflated the D_{max} compared to experimental conditions. As a result, the OLIGOMER analysis was found to produce a better fit to the experimental data than the EOM analysis.

I strongly suggest that the authors consult “Trehwella et al, 2017, 2017 publication guidelines for structural modelling of small-angle scattering data from biomolecules in solution: an update” for what is recommended to be presented for the publication of SAXS data. At the very least, fit quality should be presented in some fashion. Otherwise the results seem fairly comprehensive, and the general conclusion that there are multiple conformational states present seems supported.

We have added table 3 according to Trehwella et al.

Minor comments – there seems to be some poor sentence structure, and misuse of tense throughout the manuscript. I suggest a thorough grammar and sentence clarity check.

We have gone through the manuscript to fix the grammar and check sentence clarity.

Reviewer #2 (Remarks to the Author):

In the article “Structural basis for the mechanism of human presequence protease conformational switch and substrate recognition”, the authors utilized advanced vitrification process, the combination of self-blotting nanowire grids and piezo dispensing, during sample preparation for CryoEM and successfully obtained high-resolution structural information for three open-states Presequenc protease (PreP) together with two bound-states Prep. With further integration of size-exclusion chromatography (SEC) - small angle X-ray scattering (SAXS) and hydrogen deuterium exchange (HDX) – mass spectrometry (MS), this study proposed a model for the catalytic cycle of PreP, providing structural basis for the transition between different states of PreP as well as the selectivity and capture principle of its substrates. In addition, the comparison with the catalytic mechanism of insulin degrading enzyme (IDE) suggest a typical framework for the amyloidogenic peptide catalysis, paving the road for further tailoring the enzymatic activities. In general, this study fits the interests of the broad audience of Nature Communications, recommended for publication after addressing the comments below.

We appreciate these supportive commetns. In order to follow recommendations of Masson et al. 2019 (doi: 10.1038/s41592-019-0459-y) and provide the recommended template tables as Supplementary Tables (see Supplementary Information in doi: 10.1038/s41592-019-0459-y), we also repeated our HDX exchange experiments of PreP alone, PreP in the presence of amyloid beta, PreP in the presence of DMSO (solvent used to dissolve MB60), and PreP in the presence of MB60. Based on the new data, we revised our manuscript accordingly.

Comments:

1. In the last paragraph of the “Mechanism for the conformational switches between PreP open and closed states in the presence and absence of substrate”, the description is confusing. The expectation before HDX analysis is not unclear. Specifically, the substrate binding sites nor the binding interfaces between PreP-C and PreP-N were not mentioned in terms of residues (e.g., aa xx-xx). However, HDX protection observed in region aa 703-720, aa 892-940 are concluded as binding interfaces that meets author’s expectation. Please justify the logic and label the regions on corresponding figures.
2. Among these protected regions, aa 91-121 is not labeled nor discussed in the manuscript. Please justify the decrease of deuterium uptake upon substrate binding.

We have performed HDX-MS analysis of PreP alone, PreP+amyloid beta, PreP+DMSO (solvent control), and PreP+DMSO+MB-60 to obtain n=2 data and used that for our analysis. Using that, we have defined substrate binding sites based on King et al Structure 2014 paper prior to the discussion of the expected and unexpected HDX region induced by substrate and justify the logic accordingly. The change for aa 91-121 is not significant and it is now removed in the figure.

3. In Figure S17, the apo state of peptide 126-133 experiences a decrease in deuterium uptake at the final time point. Please explain the possible reasons.

The last point (1,000,000 second) is derived from HDX at 100,000 second at room temperature, instead of 11.6 days incubation at 0°C. This is based on the observation that room temperature has 10-fold enhanced rate than the rate at 0°C. Upon repeating, we decide to drop this sampling point since it did not provide additional information based on our data and apparently caused some confusion.

4. Substrate binding shows different structural impact on Switch A, B and C domains, among which Switch B shows a less extent HDX protection. How does this observation relate to the proposed working mechanism? Please describe.

Upon repeating the experiment, we still observed the impact of substrate binding on all three switch regions. However, we did not observe the differential effect on HDX on PreP switch A/C versus that on PreP switch B. We have revised the manuscript accordingly.

5. This paper needs better organization, especially for Figure S15 and S16. MB-60 is discussed after substrate binding, however, is shown in front.

This has been corrected.

6. Make sure the correct figures are referred in each paragraph. For example, Figure S14-16 are wrongly assigned, should be 15-17. Figure S14-15 is also wrong, should be 15-18. Please correct and check carefully throughout.

These have been corrected.

7. In the case of MB-60 binding, aa 114 – 144 experienced considerable amounts of HDX protection, however, is not discussed. Another missing peptide is aa 630-635. Please justify the observation and how can that relate to the inhibition process.

Thanks to this reviewer for so carefully going through our HDX-MS data. Upon repeating the HDX-MS, the changes of aa114-144 and aa 630-635 were minor and thus removed in figure 4 and extended data figure 19.

8. Please report key parameters for HDX analysis. For example, any smoothing functions to get the ΔD plots between different states? What is the value?

HDX-MS data were processed and analyzed using HDEaminer v2.5.1. Key parameters for HDX analysis were reported in the HDXM_MS section of Materials and Methods.

9. In the introduction, there are missing spaces in (HDX) – mass spectrometry. Please correct.

Corrected.

10. Please keep the terminology consistent throughout the manuscript, HDX or DXMS.

We have used HDX-MS throughout when appropriate.

11. The label “Figure 1” is missing in Figure 1. Please correct.

Fixed.

12. Please keep consistency on the color code of the same domain and label. In Figure 2, the protein “Switch A” should be yellow.

Fixed.

13. In Movie 5, the Switch B is mislabeled. Please correct.

Fixed.

Reviewer #3 (Remarks to the Author):

In this manuscript the authors describe a collection of structure of apo-PreP open states along with A β - and citrate synthase bound PreP at 3.3 Å-4.6 Å resolution. The authors use a combination of cryo-EM, SEC_MALS and HDMX to investigate the details of PreP catalytic cycle and conformational changes. This paper is intended to a broad audience given the importance of the subject and technical applicability of the methods used to overcome the limitations due to the small molecular weight of the complex and preferential view.

General Comment:

Given the lack of line indicators in the manuscript it is very challenging to indicate where to make edits.

Points to address:

1- The authors indicate that the use of the Spotiton dramatically improve sample preparation, particle orientation and air water interface but they lack to show tomograms for these samples. I think this panel would be extremely useful.

We do not specifically state that the use of spotiton dramatically improve sample preparation, particle orientation and air water interface in the manuscript. Rather, based on 2D and 3D classification from grids prepared using a Vitrobot, we discovered the issue of preferential denaturation of PreP-C domain. We thus hypothesized that the use of chameleon (commercial version of Spotiton) could help to reduce the denaturation. We observed that the use of chameleon eliminated the class of PreP that contained the denatured PreP-C domain. We did not specifically acquire tomograms for these datasets as we have previously acquired many, many tomograms, on both Vitrobot and Chameleon grids, and the result we always observe is that most particles are nevertheless in close proximity to the air water interface (AWI). Thus we do not expect the Chameleon grids to look much different from the Vitrobot grids when observed in a tomogram. We hypothesize that the advantage of Chameleon over the Vitrobot is the reduction of time spent at the AWI, not the complete absence of the AWI effect. In our discussion, we did propose that PreP represents a good model protein to assess the efficacy of newly emerging vitrification approaches. We have revised the cover letter to state this more precisely.

2- Can the authors comment on the unusual behavior of the FSC curves presented in Figure S8C? This is generally a behavior observed for datasets still presenting a large amount of heterogeneity. I wonder if subclassification was attempted.

We agree that this is likely due to the large amount of heterogeneity. We did attempt to subclassification but did not see distinct subclasses. Thus, we suspect that this is due to continuous heterogeneity, instead of discrete subclasses. We also attempted the use of Relion3.0 multi-body analysis to address the heterogeneity but the analysis was not successful perhaps due to the small particle size of monomeric human PreP (~110 kDa).

3- The EM data acquisition and processing have been performed expertly, but a few minor details should be addressed.

Most importantly, both samples appear to have a rather strong preferred orientation in ice. Certain regions of Euler space are overpopulated, and thus the resolution of the reconstruction is not isotropic. It is likely that both the global and local resolutions of the structures based on the FSC are inflated due to this preferred orientation. This should be addressed by reporting the anisotropy of the reconstruction by one of several reported methodologies (from the labs of Chris Russo, Dmitry Lyumkis), and the data should be reprocessed with normalization of Eulers to reduce the impact of the preferred views on the resolution.

We used 3D-Fourier Shell Correlation (FSC) algorithm described in Tan et al Nature Methods 2017 to assess orientation bias which shows that all the maps have sphericity values of 0.837 or higher. The values are now included in the table of the manuscript and the plots are included as the supplemental data for the reviewers.

4- Given the importance of the vitrification for obtaining a sample amenable for structure determination, details regarding blotting paper used.

We used the standard Vitrobot filter paper, grade 595 from Whatman. This has been added to the manuscript.

5- Since "dose" is a volumetric measurement and reported in A3, "electron exposure" (or fluence) and "exposure rate" (or flux) should be used in the methods and Table S1.

Revised.

Minor point:

1- There is a repetition in the first section of the results:” Apo-PreP grids were then prepared by Vitrobot and a dataset of 2,626 micrographs collected at 300 kV on a Titan Krios.“

Fixed.

2- Following this section there is another one for which the meaning is not necessarily clear: The two major classes, comprising ~208,000 and ~118,000 particles, displayed an intact PreP-N domain but a denatured PreP-C domain and were refined to 4.2 Å (Fig. 1B, Extended data Fig. 4,5)”. Is “but” here placed instead than “and”?

Revised.

3- In figure S5 the last two helices and structure are different. Could the rendering be made in the same way to have self-consistency within the figure?

Revised.

REVIEWERS' COMMENTS

Reviewer #1 (Remarks to the Author):

I feel the manuscript is significantly improved and recommend publishing it.

Reviewer #2 (Remarks to the Author):

The author has appropriately addressed my comments. I would now recommend Nature Communications accept this manuscript.

Reviewer #3 (Remarks to the Author):

The authors have satisfactorily revised this manuscript providing key information that was missing in the previous submission. I support the publication in this current status.

Congratulations!